# Solid-state NMR spectroscopy based atomistic view of a membrane protein unfolding pathway

Peng Xiao [1], David Bolton[1], Rachel A. Munro[1], Leonid S. Brown [1] & Vladimir Ladizhansky [1]

Membrane protein folding, structure, and function strongly depend on a cell membrane environment, yet detailed characterization of folding within a lipid bilayer is challenging. Studies of reversible unfolding yield valuable information on the energetics of folding and on the hierarchy of interactions contributing to protein stability. Here, we devise a methodology that combines hydrogen-deuterium (H/D) exchange and solid-state NMR (SSNMR) to follow membrane protein unfolding in lipid membranes at atomic resolution through detecting changes in the protein water-accessible surface, and concurrently monitoring the reversibility of unfolding. We obtain atomistic description of the reversible part of a thermally induced unfolding pathway of a seven-helical photoreceptor. The pathway is visualized through SSNMR-detected snapshots of H/D exchange patterns as a function of temperature, revealing the unfolding intermediate and its stabilizing factors. Our approach is transferable to other membrane proteins, and opens additional ways to characterize their unfolding and stabilizing interactions with atomic resolution.

[1] Department of Physics and Biophysics Interdepartmental Group, University of Guelph, 50 Stone Road E., Guelph, ON N1G 2W1, Canada. Correspondence and requests for materials should be addressed to L.S.B. (email: lebrown@uoguelph.ca) or to V.L. (email: vladizha@uoguelph.ca)

Understanding how protein amino acid sequences define their three-dimensional structures is one of the main challenges of molecular biology[1–3]. Compared to soluble proteins, folding of membrane proteins is more complex, as it requires formation of secondary structure and insertion into cell membrane, association of the secondary structure elements in a hydrophobic bilayer, and often involves oligomerization. Interactions with lipids, water, ions, other proteins, and cofactors along with intra-protein interactions, play important roles in the membrane protein folding process and contribute to their stability[3–8].

In recent years, tremendous progress has been achieved with determination of membrane protein structures, driven by advances in X-ray crystallography and cryo-electron microscopy[9], as well as by the developments in solution and solid-state nuclear magnetic resonance (SSNMR) techniques[10]. In parallel, novel experimental approaches have been developed to understand the kinetic and thermodynamic determinants of membrane protein folding and unfolding in vitro[11–18]. Membrane protein unfolding induced by external stimuli such as denaturants, pH, temperature, and mechanical forces, can be assayed by various biophysical techniques, and yields wealth of information on the hierarchy and energetics of inter-molecular and intra-molecular interactions responsible for protein stability and folding, particularly in cases where such stimulated unfolding is reversible[1,2,16,18]. Significant insights have been provided by Single Molecule Force Spectroscopy and Fluorescence spectroscopy[16,19], Mass Spectrometry coupled with isotopic exchange or covalent labeling[15], pulsed proteolysis[20], Electron Paramagnetic Resonance[21], most often in combination with site-directed mutagenesis and chemical denaturation. In particular, force spectroscopy with microsecond time resolution[11] and Hydrogen/Deuterium exchange mass spectrometry[22] showed the potential to provide nearly site-specific information about unfolding intermediates.

In the majority of membrane protein folding studies, partially unfolded states are obtained in detergents or mixed micelles and often represent structurally heterogeneous and transient ensembles which are not readily amenable to studies by high-resolution structural methods. In this contribution, we describe an approach that allows us to visualize the unfolding pathways of multi-spanning helical membrane proteins in their native-like lipid environment. Our method takes advantage of the common property shared by many lipid-embedded alpha-helical bundles: they contain a hydrophobic core which is well protected from solvent and non-exchangeable in the folded lipid-bound state. Unfolding induced by external stimuli exposes the core to water; changes in the water accessible surface can be subsequently detected site-specifically through a combination of Hydrogen/Deuterium (H/D) exchange and Magic-Angle Spinning (MAS) SSNMR.

Here, we employ thermally induced denaturation as the well-controlled external stimulus and detect H/D exchange by multi-dimensional MAS SSNMR to site-specifically follow gradual unfolding of a membrane protein as a function of increasing temperature below the irreversible denaturation point (Fig. 1). Using thermal denaturation ensures that the unfolding process of a membrane protein always occurs in a lipid-embedded environment without mixing with detergents or other denaturants, accounting for the effects of bilayer hydrophobicity, curvature, specific lipid head groups, lateral pressure, and other direct and indirect effects of lipids. Phase transition of lipids constituting most biological membranes usually occurs at low temperatures and does not overlap with the unfolding transition of helical multi-spanning membrane proteins. Nevertheless, it is well known that lipids greatly affect thermal stability of membrane proteins, increasing it compared to detergents[23–25]. Moreover, aside from general protective effect of the bilayer, some lipid species contribute to specific lipid protein interactions increasing thermal stability of membrane proteins[26–29].

We validate our method on a retinal-binding seven-helical membrane-bound photoreceptor Anabaena Sensory Rhodopsin (ASR), for which we obtain a qualitative description of the unfolding pathway and determine an atomistic map of its stabilizing interactions in the physiological lipid environment. ASR initiates a unique phototransduction cascade involving a soluble transducer which interacts with DNA and likely regulates expression of several proteins responsible for photosynthesis and the circadian clock in cyanobacterium Anabaena sp. PCC 7120[30]. ASR forms stable trimers in E.coli membranes, detergents and lipids, which can assemble into 2D lattices[31,32]. Its monomer is arranged into a seven transmembrane (TM) alpha-helical bundle[33], with an all-trans-retinal cofactor covalently bound to lysine in the seventh helix through a protonated Schiff base. We have previously obtained extensive SSNMR spectroscopic assignments for more than 90% of residues of lipid-embedded ASR (BMRB ID: 18595, Supplementary Fig. 1)[34], and solved its three-dimensional oligomeric structure[32,35]. Along with site-specific detection of residues becoming exposed to the solvent in the process of gradual unfolding, SSNMR also allows for the detection of the reversibility of unfolding at the atomic level which can be monitored through changes in the cross-peak positions and linewidths. In addition, UV-Vis spectroscopy provides an alternative, convenient means to follow the reversibility of unfolding, as irreversible denaturation in retinal-binding membrane proteins is associated with the loss of retinal chromophore[36,37] and the reduction of the corresponding absorption band. The wealth of spectroscopic and structural information available for ASR sets the stage for probing its unfolding pathways under native-like conditions.

## Results

**Thermal unfolding and H/D exchange of ASR.** Unfolding of ASR was induced by incubating the lipid-embedded ASR sample in the $D_2O$ based buffer at several temperatures in the 20–83 °C range which covers the unfolding transition (Supplementary Fig. 2), as determined by Differential Scanning Calorimetry (DSC) measurements. The lipid mixture used in this study (DMPC/DMPA, 9:1 w/w) has a transition temperature at ~25 °C as was directly determined by us previously[38], close to that of pure DMPC and much lower than the unfolding transition range of

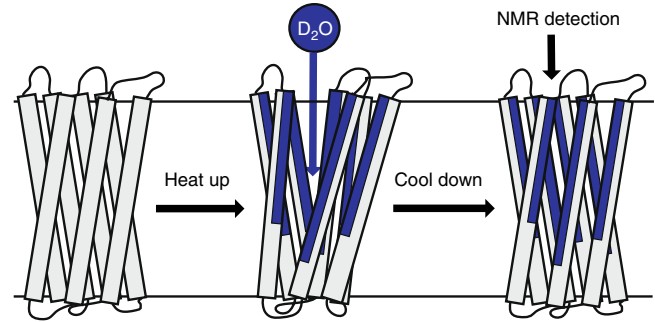

**Fig. 1** Schematics of the NMR-detected H/D exchange experiment. The sample is incubated at elevated temperatures, resulting in a gradual temperature-dependent increase of the solvent-accessible surface with amide protons at exposed sites exchanging for deuterons, and subsequently cooled down for SSNMR detection. This cycle is repeated for each elevated temperature point to form a series of NMR spectroscopic snapshots which follow the unfolding pathway. Blue color represents exchanged parts of the protein

the lipid-embedded ASR detected by DSC, ~70–85 °C (Supplementary Fig. 2), thereby indicating that the DSC transition peak arises from a large conformational change in ASR. In contrast to chemical denaturation that often uses harsh detergents, thermal unfolding in membranes allows controlling the denaturation extent of a protein without changing its native-like chemical environment. While most backbone nitrogen atoms in the TM core of ASR are protected from H/D exchange at room temperature[32], temperature-dependent unfolding gradually exposes them to the solvent. The exchange of exposed amide protons for deuterons yields the precise locations of the unfolding events. Following each incubation, the sample was cooled to 5 °C, effectively resealing the protein core, preserving the high-temperature H/D exchange pattern, and providing snapshots of partially unfolded conformations (Fig. 1). Multidimensional SSNMR spectra were taken after each incubation to determine the extent of H/D exchange, and to characterize the reversibility of the unfolding-refolding cycle.

**SSNMR spectroscopy detection of H/D exchange**. The unfolding was followed site-specifically using 2D $^{15}N$–$^{13}C\alpha$ (NCA) experiments, and 3D $^{13}C\alpha$–$^{15}N$–$^{13}C'$ (CANCO) experiments with an additional *Rotary Resonance* Recoupling (R$^3$)[39] filter (R$^3$-CANCO) (Supplementary Fig. 3). These two sets are complementary in a sense that the 2D NCA experiment detects amide protons in the non-exchangeable core, whereas the signals in the 3D R$^3$-CANCO experiment correspond to the exchangeable solvent-exposed amides. In addition, chemical shifts and widths of NMR spectral lines serve as qualitative reporters on the reversibility of unfolding.

Two independently prepared but otherwise similar ASR samples were used in this study. A larger sample (sample 1, ~10 mg of ASR) was used to record 2D NCA and 3D R$^3$-CANCO spectra, and sample 2 was used to validate our conclusions. In the 2D NCA spectroscopy (Fig. 2), a short $^1H$–$^{15}N$ cross polarization (CP)[40] excitation time of 300 µs results in NMR signals being nearly selectively excited from amide protons. Solvent exposure and H/D exchange result in significant signal intensity reduction or disappearance of peaks from exchanged residues. The contributions from dipolar spin-spin couplings between amide nitrogens and remote protons, e.g., H$_\alpha$ and H$_\beta$, remain small and are below 25% of those from amide protons, as evident from strongly attenuated signals of prolyl residues (Fig. 2a). Prolines do not carry amide protons and behave like deuterated residues, thereby serving as internal controls. Approximately 30% of residues can be unambiguously identified in the 2D NCA spectrum collected on a fully protonated ASR at 20 °C, and additional residues can be resolved when the spectra become less congested after incubation in D$_2$O at higher temperatures (Figs. 2c and 2d).

A complementary 3D R$^3$-CANCO experiment provides better spectral resolution and resolves nearly every residue out of the 206 previously assigned (Supplementary Fig. 1), and is designed to highlight the exchanged sites. The R$^3$ filter is inserted following the CA/N polarization transfer step; it recouples the dipolar interaction between the backbone $^{15}N$ and nearby protons, causing the amide signal decay. Protonated non-exchanged amide sites are affected the most, and their signals are completely suppressed. Only proline signals survive the filter in the fully protonated ASR, and serve as internal controls (Fig. 3b). The exchanged (deuterated) amide sites are affected by the R$^3$ filter to a much lesser extent (Fig. 3c). The recovery of cross-peak intensities as a function of temperature after incubation in D$_2$O-based buffer reflects gradual exchange of the exposed amides in the process of unfolding (Fig. 4), generally reaching maximal intensities when fully exchanged (e.g., Fig. 4, residue A55).

Chemical shift positions and linewidths of cross peaks are exquisitely sensitive to the local environment and represent two additional experimental observables which report on the reversibility of the unfolding-refolding cycle at the atomic level. The consistent positions and linewidths of most cross-peaks in the 2D NCA spectra after high-temperature incubation indicate that the local structure of non-exchangeable core represented by these cross peaks, remains intact and homogeneous. Small chemical shift position changes of exchangeable residues in the 3D R$^3$-CANCO spectra collected after incubation at up to 76 °C are within the range expected from isotopic effects due to deuteration of amide sites[41]. The line broadening and splitting observed for some cross peaks indicate local heterogeneity after refolding (Fig. 4).

**Thermally induced unfolding pathway derived from H/D exchange patterns**. Significant attenuation of cross-peaks intensities in the 2D NCA spectra, or reappearance of cross-peaks in the 3D R$^3$-CANCO spectra after incubation in the D$_2$O-based buffer at 20 °C (Figs. 2b, 3c, 5a) (normalized intensities below 0.25 are considered to originate from exchanged sites) mainly correspond to the solvent accessible loops and ends of helices. Most residues in the TM core and at the inter-monomer interface formed by helices B, D, and E remain protected (Fig. 6a).

We note that although solvent accessibility is expected to be the strongest factor contributing to the H/D exchange, hydrogen-bonding strength, local pH, protection of amide sites by adjacent sidechains[42], as well as the time of solvent exposure (incubation for 1 h in the 20–60 °C range, and for 2 min at above 60 °C to minimize the loss of retinal) are expected to contribute to its extent as well. Small peak attenuation (10–20%) observed for several residues in the TM regions may represent partial exchange occurring over the course of incubation. We confirmed that no significant additional exchange occurs during the 3D R$^3$-CANCO experiments (~90 h) by comparing 2D NCA spectra collected right before and after the 3D experiments.

Progression of the exchange is observed in the loops and at the ends of helices upon incubation at higher temperatures up to 76 °C, along with general signal attenuation across the entire protein (Figs. 2c, 4, 5b, Supplementary Figs. 4b–f, 5b–e). Although several residues in the TM core are exchanged in helices E, F, and G (I143, K167, T170-Y171, and S209-G212) after the incubation in the 48–68 °C temperature range, the TM core remains largely protected from solvent (Supplementary Fig 6b–e).

In agreement with this, the conserved positions and linewidths of the majority of cross-peaks in the 2D NCA and 3D R$^3$-CANCO spectra indicate insignificant structural changes up to 76 °C. $^{15}N$ chemical shift changes are mainly due to isotopic effects of deuteration of amides[41]. Line splitting and broadening observed for several residues in the 3D R$^3$-CANCO spectra (e.g., K210, P33, and G212, Fig. 4) indicate irreversible changes in both loop regions (e.g., P33) and in the TM core (K210, G212) in the vicinity of exchanged sites. However, these changes remain localized and small, less than 0.8 ppm for CO, 1 ppm for N, and 0.4 ppm for C$_\alpha$, and correspond to structurally similar substates formed after incubation at temperatures up to 76 °C.

Drastic signal attenuation is observed in the 2D NCA spectrum after incubation at 80 °C (Figs. 2d, 5c). It corresponds to a major unfolding event and is consistent with the calorimetric measurements (Supplementary Fig. 2). The conserved positions and linewidths of the remaining resonance peaks represent a structurally stable non-exchangeable core. It coexists with structurally heterogeneous (and likely locally irreversibly unfolded) parts of the protein producing broad peaks in the 2D $^{13}C$–$^{13}C$ correlation spectrum which shows both exchanged and

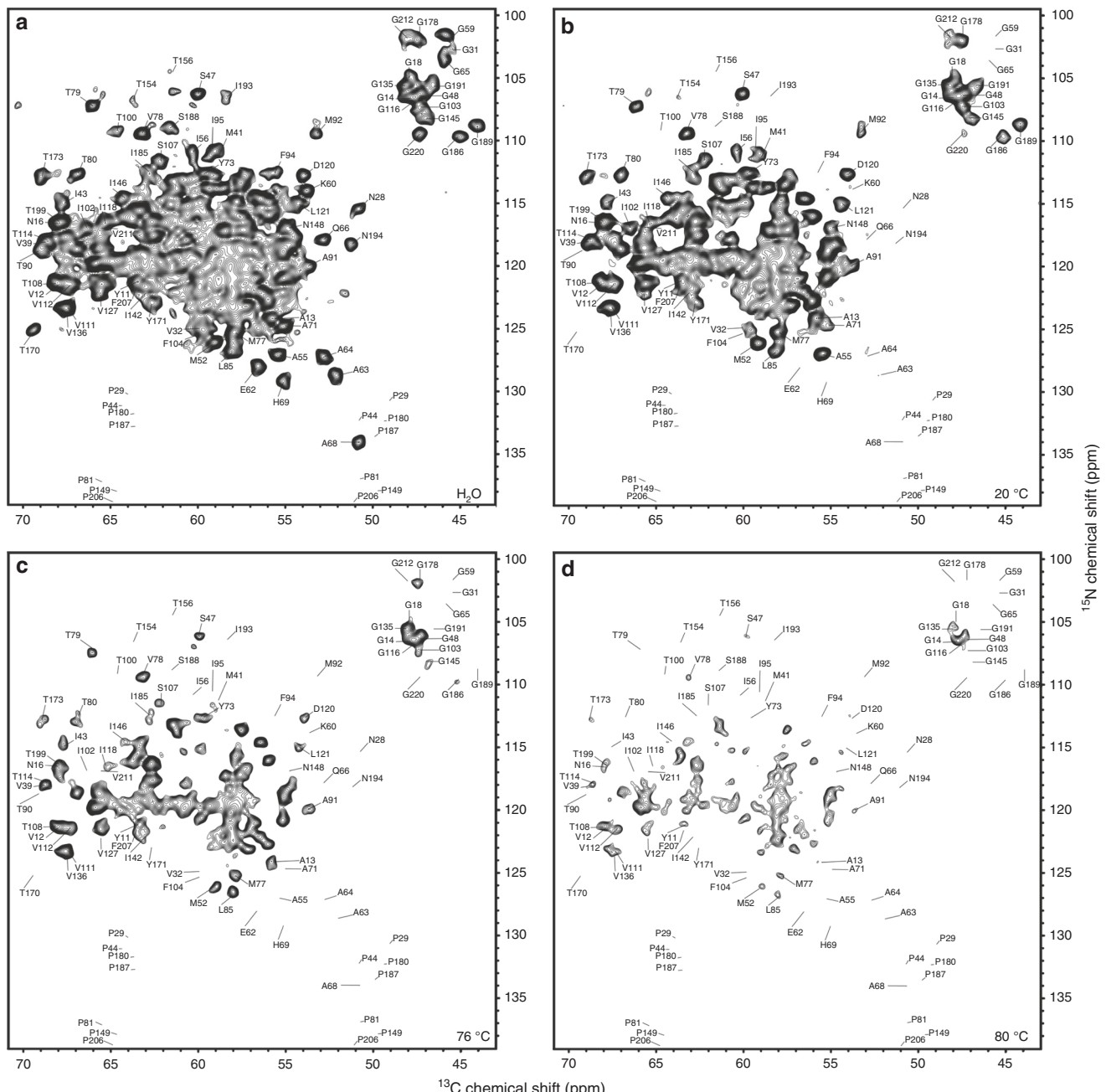

**Fig. 2** 2D NCA spectra of ASR as a function of incubation temperature. **a** Reference 2D NCA spectrum of ASR in H₂O based buffer. **b**–**d** 2D NCA spectra collected after incubation in D₂O at the indicated temperatures. Attenuated cross-peaks correspond to residues exposed to solvent due to unfolding. All spectra were collected on sample 1. The first contour in all spectra is at 5 times root-mean-square of the noise

non-exchanged residues (Supplementary Fig. 7). While large heterogeneous line broadening of most of the peaks indicates that the refolded structure represents a distribution of conformations, the overall similar shapes of the two spectra correspond to similar secondary and tertiary structures of the protein before and after the incubation at 80 °C (Supplementary Fig. 7a). Several remaining sharp peaks in the 2D NCA spectrum with conserved chemical shifts (e.g., I43, S47, V78, W131, Supplementary Fig. 7b) represent residues that do not undergo large conformational transitions. These observations are in agreement with infrared spectroscopy of ASR subjected to heating-cooling cycles similar to those used in the SSNMR experiments which shows significant backbone H/D exchange at 80 °C, as judged from the decrease of the Amide II C–N–H vibrations with concomitant increase in the

Amide II′ C–N–D band (Supplementary Fig. 8). A moderate shift of the Amide I band (backbone C=O vibrations) suggests that a significant fraction of ASR retains its secondary structure.

Absorption measurements in the visible range (Supplementary Fig. 9) indicate two protein populations after incubation at 80 °C. Approximately ~40% of ASR lose retinal, unfold irreversibly, are likely completely exchanged and "invisible" in the 2D NCA spectrum. The remaining ~60% population of retinal-bound ASR retain partially protected core which contributes to the 2D NCA spectrum (Fig. 2d), yielding the information about the partially unfolded intermediate. This distinct intermediate is formed as a result of a sharp temperature-induced transition from mostly folded state observed at 76 °C, and is followed by an unfolded state formed at 83 °C. The latter state is characterized by nearly

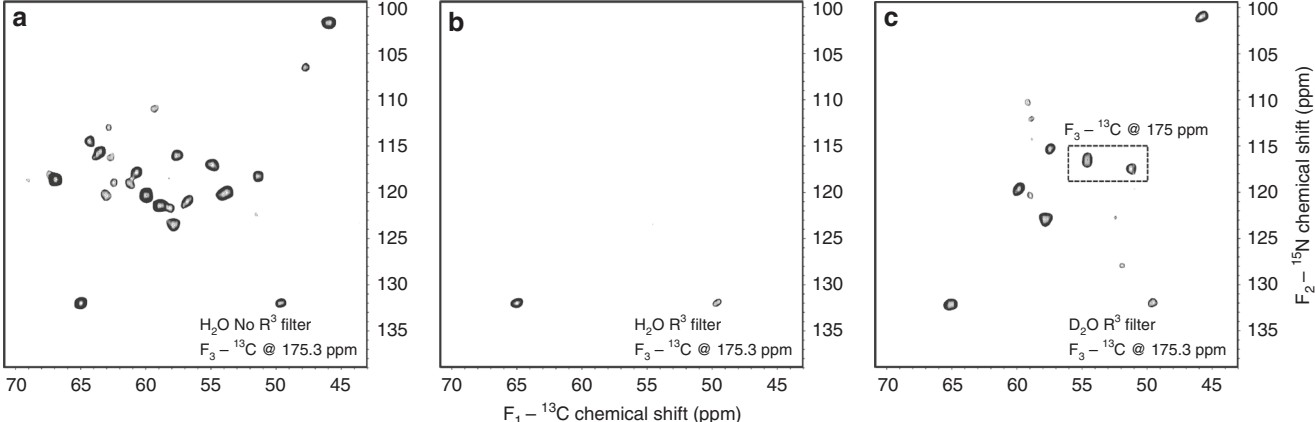

**Fig. 3** Representative 2D planes of the 3D R³-CANCO spectra. **a** Reference spectrum collected without R³ filter in the H₂O based buffer. **b** Spectrum collected with R³ filter in the H₂O based buffer. Only peaks corresponding to prolines are detected. **c** Spectrum collected with R³ filter after incubation in the D₂O based buffer at 20 °C. Several peaks corresponding to exchanged residues reappear. Residues shown in the dashed box undergo a slight isotopic shift of ~0.3 ppm in the carbonyl chemical shift dimension. All spectra were collected on sample 1. The first contour is at 5 times root-mean-square of the noise

complete exchange of the backbone amides, and by a dramatic loss of color and retinal (Supplementary Figs. 4h, 5g, 6g).

To validate our results and ensure their reproducibility, we have conducted an additional set of experiments on an independently prepared ASR sample (sample 2). Overall, similar behavior was observed for this sample (Supplementary Figs. 10, 11). ASR undergoes largely reversible unfolding after incubation at 76 °C prior to the major unfolding event, as evident from similar 2D NCA peak patterns (Supplementary Fig. 10c) and from conserved chemical shifts in the 2D ¹³C–¹³C correlation spectrum (Supplementary Fig. 11a). Unfolding becomes partially irreversible at 80 °C: the preserved, correctly folded core represented by the cross peaks in the 2D NCA spectrum is similar to that of sample 1 (Supplementary Fig. 10d), whereas irreversibly unfolded parts of the protein result in broad peaks in the 2D ¹³C–¹³C correlation spectrum (Supplementary Fig. 11b). Small discrepancies between samples 1 and 2, e.g., appearance of a cross peak corresponding to A13 in sample 2 at 80 °C, may be due to the different amount of protein in sample 2, or slight variations in sample incubation conditions.

**Structure of the partially unfolded intermediate state**. The partially unfolded state observed at 80 °C is characterized by a non-exchangeable slab approximately aligned with all-*trans* retinal cofactor; the slab spans all helices and includes most of the retinal-binding pocket (e.g., W76, F139, W176, Y179, and W183, Supplementary Fig. 12). It measures the widest in helices A, B, and E at the inter-monomer interface and narrows down to roughly one turn in helices C, F, and G (Fig. 6c). This non-exchangeable slab in the TM core is likely to play an important role as the structurally stable cluster that resists large conformational changes in the helical bundle upon thermal unfolding. Although it is the narrowest for helices in the monomer interior (e.g., helices C, F, and G), there are significantly more inter-helical side chain contacts in the slab regions of these helices. Retinal stabilizes the slab, serving as an interaction hub for several helices: in particular, it forms multiple additional non-covalent contacts with the retinal-flanking helices C and F via aromatic interactions, and contributes to the stability of the helical bundle. This suggests its key role in folding of ASR, in agreement with the results obtained for the homologous bacteriorhodopsin[43–45]. The stabilizing role of retinal is also in line with the three-stage model which postulated the importance of cofactors in membrane protein folding[4].

ASR remains a trimer at 80 °C, as most residues at the inter-monomer interface in helices B, D, and E remain non-exchanged (Fig. 6c, right panel). In contrast to the extensive exchangeability of the monomer interior, limited exchangeability of the oligomeric interface attests to its rigidity and the associated thermal stability. The rigid oligomeric interface observed at a relatively late unfolding stage suggests that the strong inter-monomer interactions are likely required to stabilize the folded state and may also be an essential driving force required for proper folding.

Helix G stands out as having the most extensive uniform exchange at 80 °C (Fig. 6c–d). The exchange gradually propagates from S209 in the vicinity of the retinal-binding K210 in the TM core towards the cytoplasmic end of the helix in the 20–76 °C range (Supplementary Fig. 6a–e), until helix G becomes completely exchanged outside of the slab at 80 °C. This likely results from a displacement or partial unwinding of helix G in 1–2 residue steps, which gradually widens a water-accessible cavity within the helical bundle. As suggested by the exchange pattern at 60 °C, the cavity propagation starts at the S209-G212/S214 stretch of helix G and involves residues T170-Y171 in helix F. This is consistent with the existence of the internal polar hydrogen-bonded network found by X-ray crystallography[33] in the cytoplasmic half of ASR, which includes several water molecules, backbone, and polar sidechains, mainly in helices C and G.

The displacement of a single helix G is consistent with the exchange patterns of other helices. Helices A, B, D, and E form the inter-monomer interface and do not undergo significant motions as evident from their limited exchangeability. The TM region of helix C remains nearly non-exchangeable in the range of 20–76 °C (Supplementary Fig. 6a–e) and shows only one-sided H/D exchange pattern at 80 °C at the cytoplasmic portion facing the interior of the monomer, whereas residues facing the interior of the trimer (Fig. 6d) remain protected. Finally, the displacement of helix G exposes the neighboring side of helix F, while leaving non-exchangeable the sides facing the monomer interior and helix E (Fig. 6d). The fact that helix G starts unfolding earlier than the other helices and undergoes more complete exchange suggests that complete formation of its secondary structure and interhelical contacts occurs in the late stages of folding at which other helices are formed and clustered around the retinal cofactor, reminiscent of another microbial rhodopsin, bacteriorhodopsin[13].

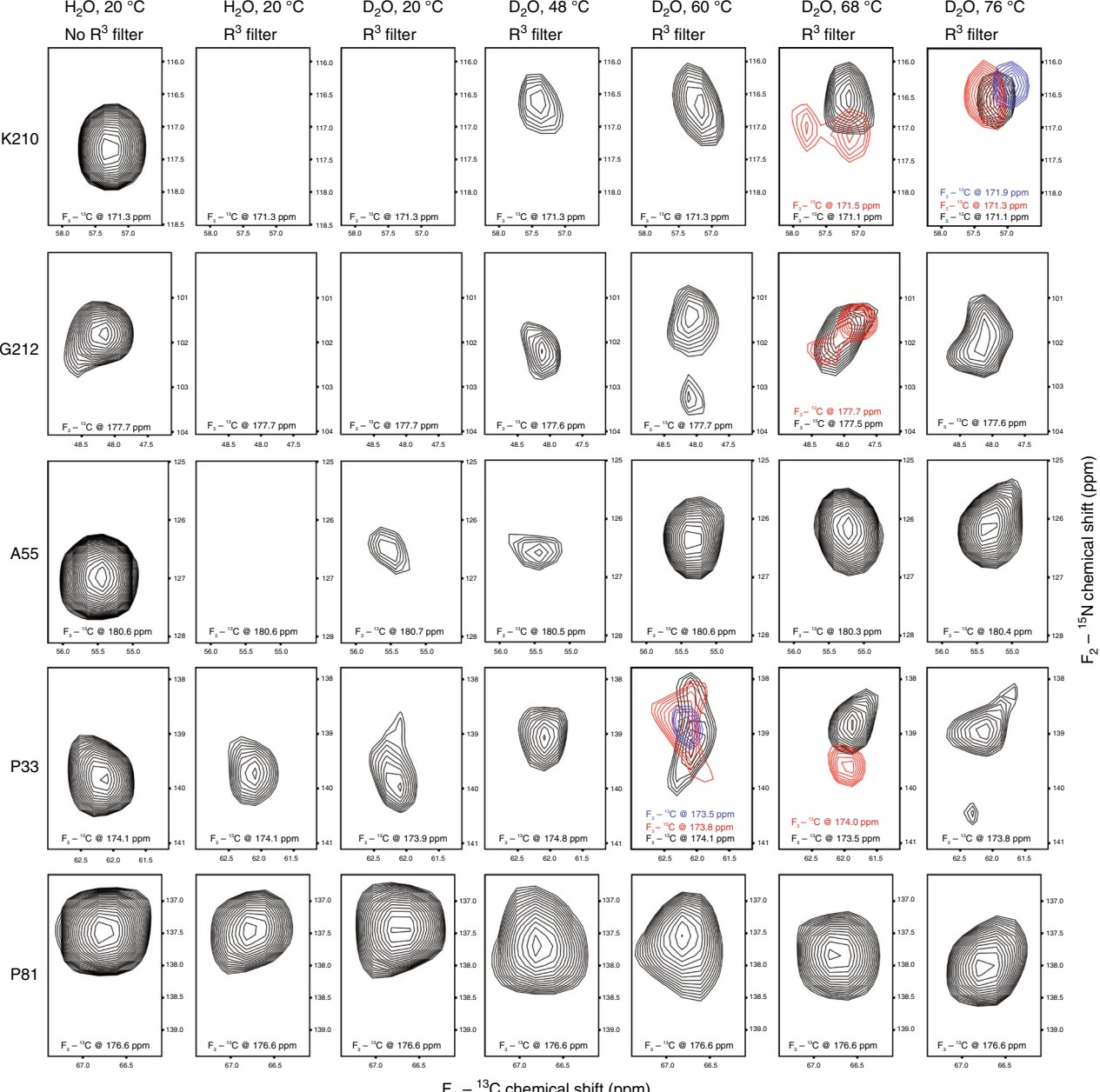

**Fig. 4** Progression of H/D exchange in 3D R³-CANCO spectra. 2D planes from selected regions of the 3D R³-CANCO spectra illustrate the effect of R³ filter and the H/D exchange as a function of incubation temperature on spectral intensities for selected residues: K210 and G212 (TM region of helix G), A55 (extracellular end of helix B), P33 (A-B loop), P81 (TM region of helix C). Splitting of the peaks (red and blue) indicate multiple local conformations resulting from the heating/cooling cycle. These conformations are however similar based on the small chemical shift differences. The first contour is at 5 times root-mean-square of the noise. Buffer conditions (H₂O or D₂O), incubation temperatures, and R³ filter conditions are indicated in the top row. All spectra were collected on sample 1

## Discussion

In summary, we presented an approach to study unfolding pathways in membrane proteins with multi-spanning topology. The unfolding was induced thermally in our experiments, and its progression as a function of increasing temperature was monitored site-specifically using solid-state NMR detected H/D exchange. Two complementary sets of SSNMR experiments, 2D NCA and 3D R³-CANCO, were carried out, allowing to separately monitor the non-exchangeable core, and the exchangeable fragments, respectively. Furthermore, 2D NCA experiments employed here can be extended to 3D NCACX experiments for a complete complementary data set of the non-exchangeable sites.

Further complemented with 2D carbon-carbon correlation spectroscopy, our approach provides an atomic-resolution view of an unfolding pathway and its reversibility. While data analysis requires high spectral resolution and the knowledge of atom-specific chemical shifts, this is a typical prerequisite for SSNMR studies. Assignment methodologies are developed and assignments are already available for many membrane proteins[46,47]. In less structurally homogeneous samples, one can envision that selective labeling or other simplifying labeling techniques can be employed in order to achieve the required site-specific resolution.

Our results demonstrate that the thermally induced denaturation combined with SSNMR and H/D exchange is a

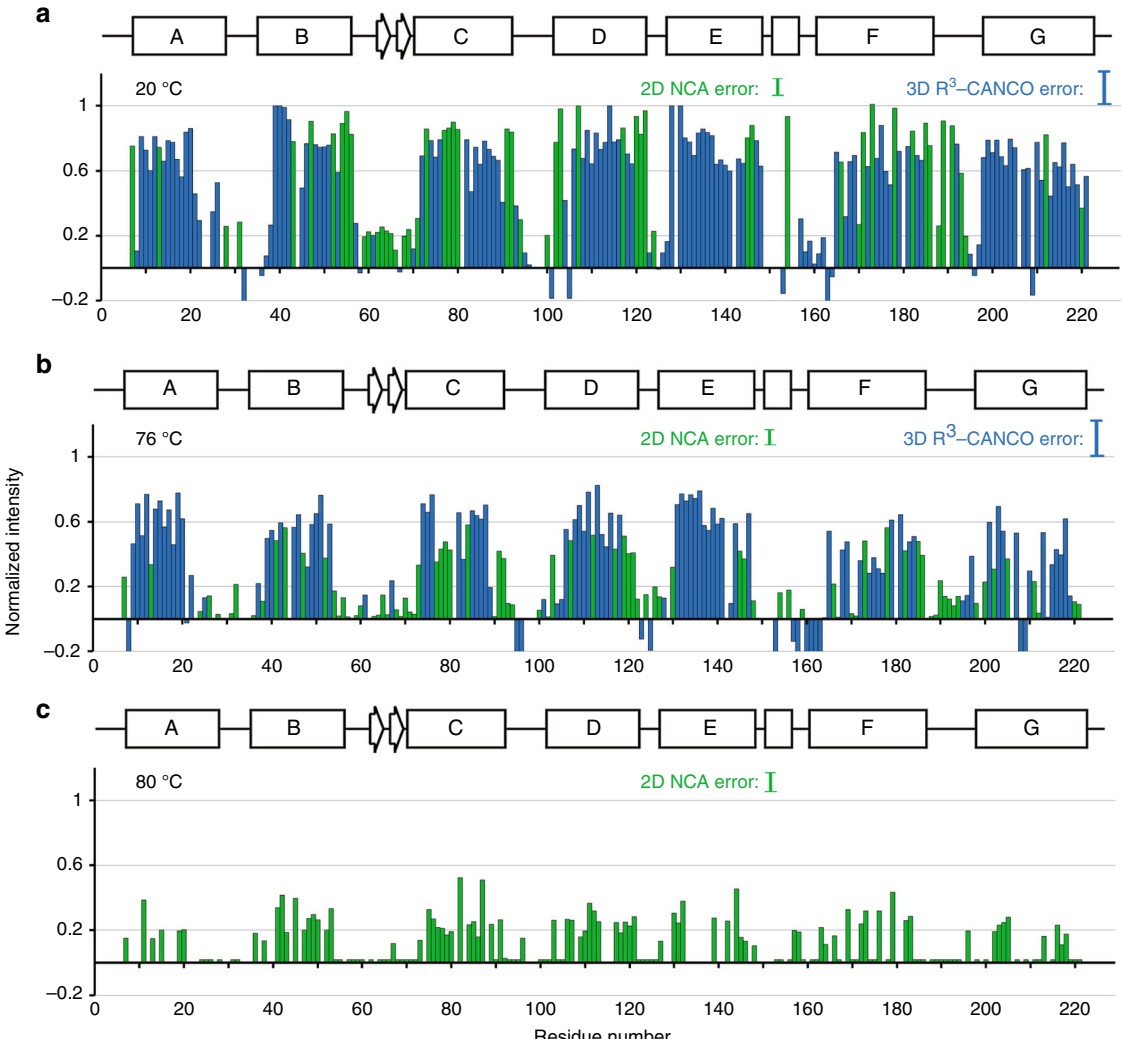

**Fig. 5** Temperature dependent progression of the H/D exchange. Normalized cross-peak intensities were obtained from the 3D $R^3$-CANCO (blue) and 2D NCA (green) spectra collected after incubation in $D_2O$ based buffer at 20 °C (**a**), 76 °C (**b**), and 80 °C (**c**). All data were extracted from spectra collected on sample 1. Normalized intensities $P_i^{NCA}$ extracted from the 2D NCA are presented as $P_i^{NCA} = S_{D_2O,i}^{NCA}/S_{H_2O,i}^{NCA}$ ratios, where $S_{D_2O,i}^{NCA}$ and $S_{H_2O,i}^{NCA}$ are peak intensities in the $D_2O$ and $H_2O$ buffers, respectively. Normalized intensities $P_i^{CANCO}$ from the 3D $R^3$-CANCO are shown as $P_i^{CANCO} = 1 - S_{D_2O,i}^{CANCO}/\left(\alpha \cdot S_{H_2O,i}^{CANCO}\right)$, where $S_{D_2O,i}^{CANCO}$, $S_{H_2O,i}^{CANCO}$ are cross-peak intensities in the $D_2O$ and $H_2O$ buffers, respectively, and $\alpha$ is a scaling factor due to the $R^3$ filter. Residues with normalized intensities lower than ~25% and negative intensities (due to normalization process described in the Methods) are considered exchanged. Errors are estimated from spectral noise, to be up to 10% in the 2D NCA and up to 20% in the 3D $R^3$-CANCO (see Methods section). Error bars are shown to guide the eye and correspond to one standard deviation for the strongest peaks

feasible approach for atomic-level structural characterization of membrane proteins unfolding under nearly physiological conditions. The experiments are conducted in a native-like lipid milieu, can be readily extended over a range of lipid compositions, pH and buffer conditions, and can include other relevant factors to evaluate their effect on protein stability. While in our implementation of the method we opted for thermal denaturation which allows for precise control of the unfolding extent by gradual temperature increase, studies of unfolding induced by other stimuli should be possible as well. Importantly, the reversibility of unfolding can be monitored at the atomic level by observing chemical shifts and widths of spectral NMR peaks.

Our approach was demonstrated using a well-studied seven-helical retinal binding membrane protein, ASR, in which we unraveled the unfolding pathway and mapped its stabilizing interactions. Unfolding begins in the TM core near retinal-binding K210, and appears to be associated with the formation of

cavity near helix G. It gradually propagates towards the cytoplasmic side with increasing temperature, until a sharp transition to a distinct intermediate occurs in the *ca*. 76–80 °C range. The unfolding intermediate state of ASR is trimeric, mainly helical (but with perturbed conformation of the seventh helix), and it retains an intact cofactor pocket. Retinal serves as an interaction hub for the interior parts of helices by forming multiple non-covalent contacts with them and contributing to the stability of the helical bundle. Oligomerization and binding of the cofactor appear to be the key factors responsible for ASR stability and may also be an essential driving force required for its proper folding. As the seventh helix G appears to be the most labile during the unfolding, one can speculate that its closing motion likely occurs at a relatively late stage of folding, in which the trimeric interface and the rest of the bundle had already formed, as was suggested for the homologous bacteriorhodopsin refolding[13]. The strong inter-monomer interactions are likely required to stabilize the folded state.

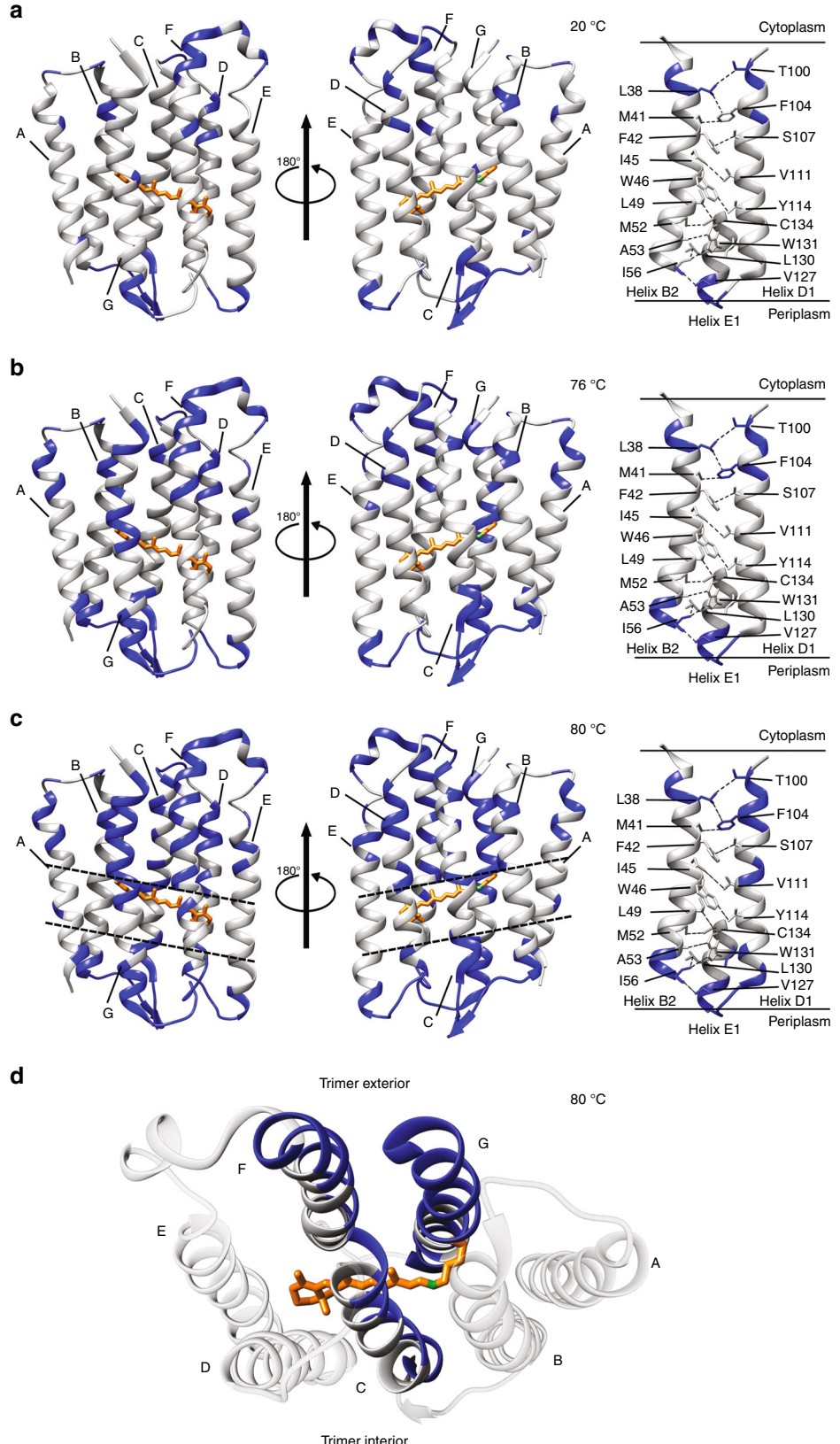

**Fig. 6** Structural representation of the H/D exchange patterns. Exchanged residues are mapped on the ASR structural model (PDB 5UK6)[35] after incubation at 20 °C (**a**), 76 °C (**b**), and 80 °C (**c** and **d**). Left panels in **a**–**c** show monomers with cytoplasmic side on top; right panels represent the inter-monomer interface. Exchanged residues are blue, non-exchangeable are gray, retinal is orange. The non-exchangeable slab region is indicated by dashed lines in **c**. **d** View from the cytoplasmic side of helices C, F, and G, highlighting the sidedness of their exchange

As our experiments were performed on the lipid-embedded protein, the uncovered hierarchy of stabilizing interactions accounts for the effect of the lipid environment. The methodology we demonstrated here is directly transferable to a broad range of other membrane proteins and complexes, and will add a powerful and precise tool to the arsenal of membrane protein folding research techniques.

## Methods

**NMR sample preparation.** Uniformly [$^{13}$C, $^{15}$N] labeled ASR sample was prepared as described previously[48]. C-terminally truncated uniformly $^{13}$C and $^{15}$N labeled His$_6$-tagged ASR (UCN ASR) was expressed in BL21 Codonplus RIL *E. coli* cells (Stratagene) grown on M9 minimal medium at 30 °C using 1 g of $^{15}$N-labeled ammonium chloride and 4 g of [U-$^{13}$C]-labeled glucose per liter as sole nitrogen and carbon sources, respectively. Protein expression was induced by the addition of IPTG to a final concentration of 1 mM when the cell density reached A$_{600}$ = 0.4 OD. Retinal was added exogenously at the time of induction at a concentration of 7.5 μM. The cells were collected by centrifugation and treated with lysozyme (0.2 mg per ml) and DNase I (2 μg per ml) in a lysis buffer (150 mM NaCl, 50 mM Tris base, 1 mM MgCl$_2$, pH 7.2) prior to being broken by sonication. The membrane fraction was solubilized in solubilization buffer (5 mM Tris base, pH 7.5) with 1% DDM (n-dodecyl β-D-maltoside) at 4 °C and purified following the batch procedure described in the Qiagen Ni$^{2+}$-NTA resin manual. The binding was carried out in a binding buffer (300 mM NaCl, 50 mM Tris base, pH 8) mixed with Ni$^{2+}$-NTA resin. The protein-bound resin was washed at least 4 times in a washing buffer (300 mM NaCl, 50 mM Tris base, 40 mM imidazole, pH 8) with 0.05% DDM, and the protein was eluted by increasing the imidazole concentration to 500 mM until the resin is completely bleached. Purified ASR from 1 L of culture was concentrated to approximately 3 ml in a reconstitution buffer (5 mM NaCl, 10 mM Tris Base, pH 8) with 0.05% DDM. Liposomes were prepared by hydrating dried DMPC and DMPA lipids mixed at a 9:1 ratio (w/w). They were mixed with the DDM-solubilized ASR and additional Triton X-100 at a Protein:Lipid:Triton ratio of 1:0.5:0.34 (w/w/w) in the reconstitution buffer and stirred for 6 h at 5 °C. Detergent was removed by adding 0.6 mg ml$^{-1}$ of Bio-beads (SM-II, Bio-Rad Laboratories, Inc., Hercules, CA, USA) and mixing for 24 h. The removal of Triton was confirmed by the disappearance of the Triton X-100 absorption bands in the FTIR spectra. Proteoliposomes were removed from Bio-beads by a 27 G syringe needle and collected by ultracentrifugation at 150,000 × g applied for 50 min. The reconstitution buffer was exchanged to a pH 9 NMR buffer (10 mM NaCl and 24 mM CHES), and the sample was further ultracentrifuged at 900,000 × g for 3 h into a small pellet which was packed into a thin-wall 3.2 mm Bruker rotor for solid-state NMR (SSNMR) measurements.

Approximately 10 mg of UCN ASR sample (sample 1) was packed for the main NMR experiments described in the Result section (Figs. 2–6, Supplementary Figs. 4–7); approximately 4 mg of a separately produced UCN ASR sample (sample 2) was packed for the additional 2D NMR experiments used to demonstrate data reproducibility (Supplementary Figs. 10, 11).

Similar protocol was followed for the preparation of lipid reconstituted ASR for DSC, FTIR, and UV-Vis measurements, except that ASR was expressed using non-isotopically labeled glucose and ammonium chloride.

**SSNMR detected H/D exchange experiments.** Thermal unfolding of ASR was achieved by incubating the sample in the D$_2$O based buffer at elevated temperatures. For each temperature point in the range of 20–83 °C, the NMR rotor (sample holder) packed with ASR was uncapped and placed in a 0.5 mL Thermowell PCR tube filled with the D$_2$O buffer (10 mM NaCl, 24 mM CHES, pD 9). This procedure ensures homogeneous hydration of the sample, as evident from complete disappearance of the signals from exchangeable residues in the 2D NCA spectra (Fig. 2b).

Sample 1 was incubated at elevated temperatures of 20 °C, 48 °C, 60 °C, 68 °C, 76 °C, 80 °C, and 83 °C, and subsequently cooled down to 5 °C for SSNMR detection after each incubation. The incubation times in the 20–60 °C range (below the pre-transition onset) were kept at 1 h to ensure complete exchange; no appreciable sample degradation happens at these temperatures. The incubation time was reduced to 2 min for higher temperatures of 68 °C, 76 °C, 80 °C, and 83 °C to minimize sample degradation as judged by the loss of retinal.

Temperature control was carried out with an Eppendorf Mastercycler Personal unit, and the experimental temperature was determined by measuring the buffer temperature immediately at the end of each incubation period using an OMEGA HH506R Digital Thermometer with Type K thermocouple. Temperature gradient across the sample during incubation was estimated by measuring temperature inside an empty rotor filled with buffer—the temperature at the center and top parts of the rotor was 3–5 °C lower than the measured buffer temperature. Such temperature gradient is expected to have negligible effect on the H/D exchange below the unfolding transition onset, but results in a distribution of states unfolded to a different extent and therefore, of the states with different extent of H/D exchange in the transition range.

**SSNMR spectroscopy.** All chemical shift correlation spectra were collected on a Bruker Avance III spectrometer operating at a magnetic field strength of 18.8 T corresponding to a proton Larmor frequency of 800.150 MHz. Bruker EFREE $^{1}$H–$^{13}$C–$^{15}$N 3.2 mm probe was used for all measurements. The spinning frequency was kept at 14.3 kHz, and temperature was 5 °C in all NMR measurements. The experimental pulse sequences are shown in Supplementary Fig. 3. Detailed experimental parameters of NMR spectroscopy experiments are given in Supplementary Table 1.

All 2D NCA experiments were recorded using a short $^{1}$H/$^{15}$N CP time of 300 μs, which ensures the polarization transfer are mainly from directly bonded protons, and minimizes the effects of dipolar couplings to remote protons. In an R$^3$-CANCO experiment, continuous irradiation with intensity matching the n = 1 R$^3$ condition[39] was applied to the proton channel following the CA/N polarization transfer step for a total duration of 839.2 μs (12 rotor cycles) to achieve a complete suppression of signals from protonated amides. The signal loss of the deuterated sites due to the R$^3$ filter from the remote two-bond $^{15}$N–$^1$H$_α$ recoupling is accounted for by a scaling factor α described in the Data Analysis below. The 2D $^{13}$C–$^{13}$C correlation spectra were recorded with 30 ms Dipolar-Assisted Rotational Resonance (DARR)[49,50] mixing.

For sample 1, a set of reference spectra consisting of a 3D CANCO (no R$^3$ filter), a R$^3$-CANCO, a 2D NCA and a 2D $^{13}$C–$^{13}$C was collected in the H$_2$O based buffer prior to the exchange experiments. 2D NCA spectra were collected for all incubation temperatures of 20 °C, 48 °C, 60 °C, 68 °C, 76 °C, 80 °C, and 83 °C, 3D R$^3$-CANCO spectra were collected for 20 °C, 48 °C, 60 °C, 68 °C, and 76 °C incubation points but not at 80 °C and 83 °C, as large inhomogeneous spectral broadening of the exchangeable sites resulted in poor spectral resolution. As an additional control of possible H/D exchange events during the 3D R$^3$-CANCO experiment (~90 h duration), we recorded 2D NCA spectra (~3 h) before and after each 3D experiment. A comparison of cross-peak intensities showed no appreciable H/D exchange during the 3D experiments. An additional 2D $^{13}$C–$^{13}$C spectrum was collected after the H/D exchange at 80 °C.

Sample 2 was used to validate our conclusions and confirm data reproducibility: 2D NCA and 2D $^{13}$C–$^{13}$C reference spectra were collected in the H$_2$O based buffer; the same H/D exchange temperature incubation protocol was followed as for sample 1, but 2D NCA spectra were collected only after the incubation at 20 °C, 76 °C, and 80 °C. Additional 2D $^{13}$C–$^{13}$C spectra were collected after the exchange at 20 °C, 76 °C, and 80 °C to monitor the reversibility of unfolding.

**Differential scanning calorimetry.** The DSC experiments were performed on two independently prepared ASR proteoliposome suspensions (~1 mg mL$^{-1}$) in the D$_2$O based NMR buffer, using TA Nano DSC 602000.901 unit, in a temperature ranges of 40–100 °C scanned at a rate of 1 °C per min. The midpoint temperature of the unfolding transition $T_m$ was found to be in the ~77–78 °C range ($ΔT_{1/2}$ = 5–6 °C, between 72 °C and 80 °C), with a pre-transition onset $T_{onset}$ at ~62 °C and the post-transition $T_{end}$ at ~89 °C (Supplementary Fig. 2). The total calorimetric transition enthalpy $ΔH_{cal}$ was estimated to be approximately −65 kJ mol$^{-1}$. Small differences between the DSC curves collected on two samples are related to small variation in protein-to-lipid ratios[26].

**FTIR spectroscopy.** FTIR measurements of ASR proteoliposomes (Supplementary Fig. 8) were conducted using a temperature-controlled Germanium Attenuated Total Reflectance (ATR) accessory (Pike Technologies, Madison WI) installed in a Vertex 70 FTIR spectrometer (Bruker, Milton ON). One hundred individual spectra were averaged at 4 cm$^{-1}$ resolution, using a DTGS detector and transmission spectra of the empty ATR accessory as references. First, the control spectrum of protonated ASR was taken at 30 °C by drying 20 μL of the proteoliposome water suspension on the surface of the germanium crystal using dry nitrogen gas. After taking the control spectrum, the film was rehydrated with 50 μL of the D$_2$O buffer, incubated for 2 min at the desired temperature, cooled back to 30 °C, the buffer was withdrawn by capillary forces, and the film was re-dried to observe the secondary structure and the extent of H/D exchange of ASR at each temperature.

**UV-VIS spectroscopy.** UV-Vis spectroscopy of ASR proteoliposome suspensions was performed using Cary-50 spectrometer (Varian). Approximately 1/10th of the ASR SSNMR sample was homogenized and suspended in 1 mL of the NMR buffer in a quartz cuvette and kept in the dark for 1 h before the absorption measurement. Each spectrum was corrected by subtracting the scattering baseline according to

$$\left( \frac{\lambda^{-0.15} - \lambda_{max}^{-0.15}}{\lambda_{min}^{-0.15} - \lambda_{max}^{-0.15}} \right)^{2.7} \tag{1}$$

where λ denotes the wavelength, and $\lambda_{min}$ and $\lambda_{max}$ are the wavelengths at the lower end and the higher end of the spectrum, respectively.

The reference spectrum (Supplementary Fig. 9, black curve) was measured in the H$_2$O based NMR buffer at 20 °C. To mimic the thermal unfolding conditions used in the NMR experiments, the sample was centrifuged and packed into a thin-walled 3.2 mm NMR rotor, incubated in a D$_2$O based NMR buffer at 80 °C for 2 min and subsequently cooled down to 20 °C, unpacked and ~1/10th of the sample

was transferred to quartz cuvette for absorption measurement (Supplementary Fig. 9, red curve). To estimate the extent of retinal loss upon heating, the two spectra were normalized in the following way:

$$A_{550nm,20°C} = A_{550nm,80°C} + A_{380nm,80°C} \cdot \frac{\varepsilon_{550nm}}{\varepsilon_{380nm}} \qquad (2)$$

where $A_{550nm,20°C}$ and $A_{550nm,80°C}$ are the absorbance amplitudes of retinal-bound dark-adapted ASR before and after incubation at 80 °C, respectively; $A_{380nm,80°C}$ is the absorbance of free retinal at 20 °C after the incubation at 80 °C; $\varepsilon_{550nm}$ and $\varepsilon_{380nm}$ are the extinction coefficients of dark-adapted ASR (55,000 $M^{-1}cm^{-1}$) and free retinal (44,000 $M^{-1}cm^{-1}$)[51], respectively.

**NMR data processing.** Carbon and nitrogen chemical shifts were indirectly referenced to DSS (2,2-Dimethyl-2-silapentane-5-sulfonic acid) by adjusting the shift of $^{13}C$ adamantane downfield peak to 40.48 ppm[52]. All spectra were processed using NMRpipe[53] and analyzed using CARA[54]. Both 2D NCA and 3D R³-CANCO data were processed with a Lorentzian-to-Gaussian apodization function; the 2D $^{13}C$–$^{13}C$ spectra were apodized using a squared cosine function. The chemical shift assignments of ASR reported in a previous study (BMRB ID: 18595)[34] were used in the data analysis.

**H/D exchange data analysis.** Deuteration of amide sites results in the attenuation of cross-peaks in the 2D NCA spectra. Signal to noise ratios (SNR) of resolved peaks were extracted from the 2D NCA spectra and normalized with respect to the reference spectrum collected under the same NMR conditions but in the $H_2O$ buffer:

$$P_i^{NCA} = \frac{S_{D_2O,i}^{NCA}}{S_{H_2O,i}^{NCA}} \qquad (3)$$

here, $P_i^{NCA}$ denotes normalized cross-peak intensity for $i$-th residue detected in the 2D NCA, $S_{D_2O,i}^{NCA}$ is a cross-peak intensity of the same residue after incubation in the $D_2O$ buffer, and $S_{H_2O,i}^{NCA}$ is a reference signal intensity of the same residue in the $H_2O$ buffer.

In contrast to 2D NCA, deuteration of amide sites results in the increase of cross-peak intensities in the R³-CANCO spectra. To make them directly comparable with those from the 2D NCA spectra, the R³-CANCO intensities were normalized according to:

$$P_i^{CANCO} = 1 - \frac{S_{D_2O,i}^{CANCO}}{\alpha \cdot S_{H_2O,i}^{CANCO}} \qquad (4)$$

here, $P_i^{CANCO}$ denotes normalized cross-peak intensity for $i$-th residue detected in the R³-CANCO, $S_{D_2O,i}^{CANCO}$ is a cross-peak intensity of the same residue after incubation in the $D_2O$ buffer, $S_{H_2O,i}^{CANCO}$ is a reference signal intensity without the R³ filter, and $\alpha$ is the scaling factor that accounts for signal losses due to the R³ filter, which is estimated to be 0.50 ± 0.07 by comparing the average SNRs of proline cross-peak intensities in the CANCO spectra with and without the R³ filter.

If resolved, relative cross-peak intensities extracted from the 2D NCA spectra (Eq. (3)) were used to represent the extent of exchange in Fig. 5 and Supplementary Fig. 5. For peaks overlapped in the 2D NCA, 3D R³-CANCO data were used for the determination of the exchange (Eq. (4)).

The progression of H/D exchange results in simplification of the 2D NCA spectra and improves its resolution, allowing for more peaks to be extracted from the 2D spectra at higher temperatures. In particular, only 2D NCA spectra were collected after incubation at 80 °C, and 83 °C, and relative cross-peak intensities were estimated from these spectra.

Errors in the determination of the normalized peak intensities extracted from the 2D NCA spectra are dominated by random contributions, and were estimated from spectral noise to be up to 10%. The uncertainty of normalized peak intensities estimated from the 3D R³-CANCO experiment contains random contributions, but is also dependent on the uncertainty of the scaling factor $\alpha$ (Eq. (4)). The latter was estimated from the analysis of the distribution of proline cross peak intensities to be ~14%, and the total error was estimated to be up to 20%.

H/D exchange pattern was visualized and mapped on a structural model of lipid-embedded ASR determined by solid-state NMR and refined using Double Electron-Electron Resonance spectroscopy (PDB 5UK6)[35] using UCSF Chimera[55]. Proline signals were used as internal controls to estimate the residual signal intensities from the deuterated sites, which were found to be ~25%. Below the major unfolding transition in the 20–76 °C range, a residue was considered to be fully exchanged and mapped on the structure in Fig. 6 and Supplementary Fig. 6 when its relative intensity was below ~0.25.

A different cutoff criterion was used to interpret cross-peak intensities after incubation at 80 °C. Two ASR populations are observed at this temperature, corresponding to the retinal-bound (~60%) and retinal-free (~40%) forms. The retinal-free population undergoes irreversible unfolding, is completely exchanged and invisible in the 2D NCA spectrum. Only the refolded retinal-bound form contributed to the 2D NCA spectrum. To compensate for the distribution of the unfolded states, the 0.2 cutoff level was empirically chosen, based on the highest

residual intensities exhibited by residues that had been determined to be fully exchanged at lower temperatures, e.g., Q157 and T196.

**Reporting summary.** Further information on research design is available in the Nature Research Reporting Summary linked to this article.

## Data availability
Other data are available from the corresponding authors upon reasonable request.

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

## Acknowledgements

This research was funded by Natural Science and Engineering Research Council of Canada (Discovery Grants RGPIN-2014-04547 to V.L., RGPIN-2018-04397 to L.S.B.), Canada Foundation for Innovation and Ontario Ministry of Economic Development and Innovation. We thank Dr. Lars Konermann (Western University) and Dr. Sameer Al-Abdul-Wahid (University of Guelph) for useful discussions.

## Author contributions

P.X., L.S.B. and V.L. designed the study; D.B., R.A.M. and P.X. prepared the samples; P.X. and V.L. conducted the NMR experiments; P.X. conducted and analyzed the DSC measurements; L.S.B. conducted and analyzed the FTIR measurements; P.X. analyzed the NMR data; P.X., L.S.B. and V.L. wrote the paper. All authors discussed the results of the study.

## Additional information

**Competing interests:** The authors declare no competing interests.

