## [Peer Review File · Nature Communications]

Reviewers' comments:

Reviewer #1 (Remarks to the Author):

This manuscript describes a technique to use thermal denaturation coupled with H/D exchange to follow the unfolding pathways of membrane proteins using solid-state NMR. Unfolded residues readily undergo H/D exchange, therefore losing their signals in ^{15}N -based 2D and 3D correlation spectra. The authors used the 7-transmembrane helix protein, Anabaena sensory rhodopsin, to demonstrate this approach. The protein is in fact relatively thermal stable, and even at $\sim 80^\circ\text{C}$ does not completely denature, but retain a structured core. The authors show that the unfolding is reversible. Because the authors have previously assigned the vast majority of the resonances of this protein, and the thermal treatment does not change the structure of the remaining core of the protein, the approach of following the remaining signals after heating, which appear at the same chemical shifts as the protein without the thermal treatment, works well.

The experiments in this study are well conducted and the data are correctly analyzed. For some proteins this approach will work as a way to investigate membrane protein unfolding. However, the applicability of this approach will likely be limited, because most biologically interesting membrane proteins will not be so ordered as ASR, but will have far more conformational motions and potentially multiple unfolding pathways. So this study, while technically sound, seems largely an exercise of resonance assignment of a protein that is chosen more for its nice NMR spectra than for an important biological question. I encourage the authors to address a truly significant membrane protein unfolding question in the future, using this or other related solid-state NMR approaches.

Reviewer #2 (Remarks to the Author):

Comments for paper titled with "Solid-state NMR spectroscopy based atomic view of a membrane protein unfolding pathway".

This paper describes detailed characterization of membrane protein unfolding within a lipid bilayer. Authors applied a new methodology that combines hydrogen-deuterium (H/D) exchange and solid-state NMR (SSNMR), to follow membrane protein unfolding in lipid membrane at atomic resolution through detecting changes in the protein water-accessible surface, and concurrently monitoring reversibility of unfolding. Authors obtained atomistic description of a thermally induced unfolding pathways of a seven-helical trimeric photoreceptor. This study opens new ways to determine membrane proteins unfolding pathways and characterize stabilizing interaction of membrane proteins with atomic resolution.

This is a careful piece of work to reveal unfolding pathways of membrane proteins through H/D exchange in atomic resolution using SSNMR spectroscopy. Therefore, this paper would be potentially published as an article of Nature Communication after some comments (see below) are carefully considered.

Comments for authors

1) Page 4, Introduction: "Here, we employ thermally induced denaturation as the well-controlled external stimulus, and detected H/D exchange by multi-dimensional MAS SSNMR to follow gradual unfolding of a membrane protein as a function of increasing temperature below the irreversible denaturation point (Fig. 1)."

The meaning of "irreversible denaturation point" is not clearly explained in introduction section and in Fig.1. Authors should give the meaning of irreversible denaturation point for the case of retinal binding membrane proteins in relation to the unfolding pathways.

2) Figure 4, Introduction: "Using thermal denaturation ensures that the unfolding process of a membrane protein always occurs in a lipid-embedded environment without mixing with detergents or other denaturants, accounting for the effects of bilayer hydrophobicity, curvature, specific lipid

head group, lateral pressure, and other direct and indirect effects of lipids.”

Author pointed out that lipid membrane play important role in the unfolding process of a membrane protein. However, authors do not discuss on the effect of lipid membrane on the unfolding of membrane proteins such as phase transition of lipid membrane from gel phase or liquid crystalline phase to isotropic phase, and also lipid protein interaction in the introduction section. This reviewer suggest that author briefly give effect of lipid membrane with unfolding pathways of membrane proteins during heating processes which may describe in introduction section.

3) Page 6, Results: “Unfolding of ASR was induced by incubating the lipid-embedded ASR sample in the D2O based buffer at several temperatures in the 20 – 83 °C range which covers the unfolding transition (Supplementary Fig. 3).”

Author imply that the DSC transition peak at 78 °C is attributed to the unfolding transition of the lipid-embedded ASR. In relation to the comment 2), this reviewer suggest that authors should mention that the DSC transition peak is not due to the phase transition of lipid membrane but probably due to large conformational change of membrane proteins possibly by releasing retinal from the proteins.

4) Page 10, Figure 3: This reviewer suggest that authors add one column (H2O, 20 °C, R3 Filter for K210, G212, A55, P33 and P81) to Figure 3d. It would be helpful for reader to understand the effect of H/D exchange in 3D R3-CANCO experiments

5) Page 11, Results: “Small chemical shift position changes of exchangeable residues in the 3D R3-CANCO spectra collected after incubation at up to 76 °C are within the range expected from isotopic effects due to deuteration of amide sites.”

In Fig. 3d, cross peak patterns of K210 and P33 at 68 °C are different from those at 76 °C. Author should also give explanation on the reason of the differences.

6) Page 13, Figure 5: Retinal position is difficult to see, it would be better show the thicker bond for retinal in the figure.

7) Page 15, Results: “The remaining ~60% population of retinal-bound ASR retinal partially protected core the reversibly unfolded state, which contribute to the 2D NCA spectrum (Fig. 2d) yielding the information about the partially unfolded intermediate.

It is not clear the meaning of “partially unfolded intermediate”. Is that same as “reversibly unfolded state” and why this state is called intermediate?

8) Page 15, Results: “Finally, incubation at 83 °C is accompanied by a dramatic loss of color, and results in nearly complete exchange of the backbone amides (Supplementary Fig. 4h, 5g, 6g).”

It is not clear weather this state after incubation at 83 °C is reversibly unfolded state as same as those appeared ~40% after incubate at 80 °C.

9) Page 19, discussion: “ASR, in which we unrevealed the unfolding pathway and mapped its stabilizing interaction. The observed unfolding intermediate state of ASR is trimer mainly helical (but with perturbed conformation of the seventh helix), and retains an intact cofactor pocket. Retinal serves as an interaction hub for the interior parts of helices by forming multiple non-covalent contacts with them and contributing to the stability of the helical bundle.”

In this discussion section, author may discuss more detailed unfolding pathways by including reversibly unfolding pathways after incubation temperature up to 76 °C, and irreversibly unfolding pathways which accompanied with losing retinal after incubation at 80 °C.

Reviewer #3 (Remarks to the Author):

Xiao et al present a comprehensive study of the thermal unfolding pathway of Anabaena Sensory Rhodopsin, a 7-helical integral membrane protein that binds retinal and forms trimers in membranes. The approach is measuring hydrogen/deuterium exchange by solid-state NMR which can be performed in a lipid bilayer environment. More than 90% of all residues of the protein have been assigned previously and a structure has also been solved previously so that the fate of a very large number of residues could be individually followed by this approach, making the study very comprehensive and including much detail of the pathway at the atomistic level. The NMR approach includes complementary NCA experiments and rotational resonance recoupling CANCO experiments following non-H/D-exchangeable and exchangeable amide protons, respectively. The transmembrane helices were found to gradually unfold from both ends as temperature was raised, leaving behind a small core slightly below the middle of the membrane that never unfolds even at 80C. The study provides much interesting new insight, but also contains some areas that need to be improved in a revision.

1. The authors emphasize much, rightfully so, the importance of reversibility in order to determine the stability of proteins, which particularly for membrane proteins represents a huge challenge. Contrary to what one might expect from the abstract and introduction, they have not demonstrated reversibility beyond the inflection point of the thermal denaturation curve which happens at 78C. That many features of the spectra might be reversible below this point, where the protein is still relatively stable, is good and should be expected, but it does not prove reversibility of the actual unfolding reaction. This point must be clarified in the text.

2. It would help much with the general understanding if the authors indicated in the quite important supplementary figure 2 with vertical lines the temperature points where their main NMR data were collected, i.e. 76 and 80 degrees for main figures and 48, 60, 68, 76, and 83 degrees for supplementary figures. BTW, the data of supplementary figures 4, 5, and 6 should be better referenced in the main text.

3. According to Online Methods, all experiments were performed on a single sample. Although I realize that these types of experiments are demanding, demonstrating reproducibility is important, both for the NMR and DSC data. Some key experiments should be repeated with an independently prepared sample.

4. Related to the previous point, the lipids for bilayer reconstitution were DMPC:DMPA (9:1), but there was also some TX100 detergent needed in the reconstitution process. Was the detergent fully removed by 24h Bio-bead treatment? How much might have been left in the sample and how could this affect the unfolding data? It is important to show that the sample preparation procedure and results are reproducible and therefore support the overall conclusions of the paper.

5. (minor) Some of the figure panels are rather small (e.g. figure 3a-c) and hard to read.

Response to Reviewers' comments.

Reviewer #1:

This manuscript describes a technique to use thermal denaturation coupled with H/D exchange to follow the unfolding pathways of membrane proteins using solid-state NMR. Unfolded residues readily undergo H/D exchange, therefore losing their signals in ^{15}N -based 2D and 3D correlation spectra. The authors used the 7-transmembrane helix protein, Anabaena sensory rhodopsin, to demonstrate this approach. The protein is in fact relatively thermal stable, and even at $\sim 80^\circ\text{C}$ does not completely denature, but retain a structured core. The authors show that the unfolding is reversible. Because the authors have previously assigned the vast majority of the resonances of this protein, and the thermal treatment does not change the structure of the remaining core of the protein, the approach of following the remaining signals after heating, which appear at the same chemical shifts as the protein without the thermal treatment, works well.

The experiments in this study are well conducted and the data are correctly analyzed. For some proteins this approach will work as a way to investigate membrane protein unfolding. However, the applicability of this approach will likely be limited, because most biologically interesting membrane proteins will not be so ordered as ASR, but will have far more conformational motions and potentially multiple unfolding pathways. So this study, while technically sound, seems largely an exercise of resonance assignment of a protein that is chosen more for its nice NMR spectra than for an important biological question. I encourage the authors to address a truly significant membrane protein unfolding question in the future, using this or other related solid-state NMR approaches.

Response: We agree that applying our methodology to proteins with more complex unfolding landscape would be an exciting application. In fact, we already started collecting preliminary data on a membrane protein of different membrane topology, human aquaporin 1, a ubiquitous water channel present in multiple human cells and tissues and implicated in various pathologies. The approach described in this paper appears to work well for human aquaporin 1 (data analysis is still in progress), and it will be a subject of future publication.

We also agree with the reviewer that for some membrane proteins applying our approach in its current form will be challenging, mainly due to the lack of (or difficulties in obtaining) resonance assignments. We believe that even in these cases our approach can work after some modifications, for example using selective isotopic labeling or other methods of spectral simplification. The presence of multiple unfolding pathways is not necessarily an impediment to our approach as long as they are reversible. We will detect cumulative changes in solvent accessibility for all of these pathways, but would be still able to identify the protected core. We have revised our discussion on page 15 as follows:

“While data analysis requires high spectral resolution and the knowledge of atom-specific chemical shifts, this is a typical prerequisite for SSNMR studies. Assignment methodologies

are developed and assignments are already available for many membrane proteins^{46,47}. In less structurally homogeneous samples, one can envision that selective labeling or other simplifying labeling techniques can be employed in order to achieve the required site-specific resolution.

Reviewer #2

1) Page 4, Introduction: *“Here, we employ thermally induced denaturation as the well-controlled external stimulus, and detected H/D exchange by multi-dimensional MAS SSNMR to follow gradual unfolding of a membrane protein as a function of increasing temperature below the irreversible denaturation point (Fig. 1).”*

The meaning of “irreversible denaturation point” is not clearly explained in introduction section and in Fig.1. Authors should give the meaning of irreversible denaturation point for the case of retinal binding membrane proteins in relation to the unfolding pathways.

Response: Thank you for the comment. The irreversible denaturation point refers to a temperature point at which the protein unfolds and does not return to its original conformation, typically leading to a structurally heterogeneous state and broad NMR spectra upon cooling. Reversibility can be monitored through chemical shift perturbations and line broadening and independently by UV-Visible spectroscopy. We clarified this in the text (page 5-6):

“Along with site-specific detection of residues becoming exposed to the solvent in the process of gradual unfolding, SSNMR also allows for the detection of the reversibility of unfolding at the atomic level which can be monitored through changes in the cross-peak positions and linewidths. Additionally, UV-Vis spectroscopy provides an alternative, convenient means to follow the reversibility of unfolding, as irreversible denaturation in retinal-binding membrane proteins is associated with the loss of retinal chromophore^{36,37} and the reduction of the corresponding absorption band.”

2) Figure 4, Introduction: *“Using thermal denaturation ensures that the unfolding process of a membrane protein always occurs in a lipid-embedded environment without mixing with detergents or other denaturants, accounting for the effects of bilayer hydrophobicity, curvature, specific lipid head group, lateral pressure, and other direct and indirect effects of lipids.”*

Author pointed out that lipid membrane play important role in the unfolding process of a membrane protein. However, authors do not discuss on the effect of lipid membrane on the unfolding of membrane proteins such as phase transition of lipid membrane from gel phase or liquid crystalline phase to isotropic phase, and also lipid protein interaction in the introduction section. This reviewer suggest that author briefly give effect of lipid membrane with unfolding pathways of membrane proteins during heating processes which may describe in introduction section.

Response: We added a brief discussion of these trends into the introduction on pages 4-5:

“Phase transition of lipids constituting most biological membranes usually occurs at low

temperatures and does not overlap with the unfolding transition of helical multi-spanning membrane proteins. Nevertheless, it is well known that lipids greatly affect thermal stability of membrane proteins, increasing it compared to detergents²³⁻²⁵. Moreover, aside from general protective effect of the bilayer, some lipid species contribute to specific lipid protein interactions increasing thermal stability of membrane proteins.²⁶⁻²⁹

3) Page 6, Results: “Unfolding of ASR was induced by incubating the lipid-embedded ASR sample in the D₂O based buffer at several temperatures in the 20 – 83 °C range which covers the unfolding transition (Supplementary Fig. 3).”

Author imply that the DSC transition peak at 78 °C is attributed to the unfolding transition of the lipid-embedded ASR. In relation to the comment 2), this reviewer suggest that authors should mention that the DSC transition peak is not due to the phase transition of lipid membrane but probably due to large conformational change of membrane proteins possibly by releasing retinal from the proteins.

Response: The lipid mixture used in this research (DMPC/DMPA in 9:1 ratio) has a melting point of ~25 °C, close to that of pure DMPC, and much lower than the main unfolding transition temperature of the lipid-embedded ASR, ~77-78 °C. The DSC transition peak of ASR arises from a large conformational change of the membrane protein accompanied by a retinal release from the protein. We clarified this on page 6 (changes underlined):

“Unfolding of ASR was induced by incubating the lipid-embedded ASR sample in the D₂O based buffer at several temperatures in the 20 - 83 °C range which covers the unfolding transition (**Supplementary Fig. 2**), as determined by Differential Scanning Calorimetry (DSC) measurements. The lipid mixture used in this study (DMPC/DMPA, 9:1 w/w) has a transition temperature at ~25 °C as was directly determined by us previously³⁸, close to that of pure DMPC and much lower than the unfolding transition range of the lipid-embedded ASR detected by DSC, ~70- 85 °C (Supplementary Fig. 2), thereby indicating that the DSC transition peak arises from a large conformational change in ASR.

4) Page 10, Figure 3: This reviewer suggest that authors add one column (H₂O, 20 °C, R3 Filter for K210, G212, A55, P33 and P81) to Figure 3d. It would be helpful for reader to understand the effect of H/D exchange in 3D R3-CANCO experiments

Response: We have added the column. To improve legibility, we have split figure 3 into two figures. Figure 3 (new version) illustrates the general effect of R³ filter, whereas figure 4 shows parts of the spectra corresponding to K210, G212, A55, P33 and P81, and contains one additional column corresponding to the 20 °C incubation temperature.

5) Page 11, Results: “Small chemical shift position changes of exchangeable residues in the 3D R3-CANCO spectra collected after incubation at up to 76 °C are within the range expected from isotopic effects due to deuteration of amide sites.”

In Fig. 3d, cross peak patterns of K210 and P33 at 68 °C are different from those at 76 °C. Author should also give explanation on the reason of the differences.

Response: Cross peaks of both residues indicate local heterogeneity induced by heating. P33 is located in the AB loop and appears to refold into a different conformation after incubation at 60 °C or higher. Heterogeneity observed for K210 may arise from the local internal unfolding which begins in the immediate vicinity of K210 (e.g, S209-G212/S214). We note, however, that all chemical shift changes and splitting remain small, indicating multiple but structurally similar substates. We have revised our discussion on page 10 as follows (changes underlined):

“In agreement with this, the conserved positions and line widths of the majority of cross-peaks in the 2D NCA and 3D R³-CANCO spectra indicate insignificant structural changes up to 76 °C. ¹⁵N chemical shift changes are mainly due to isotopic effects of deuteration of amides.⁴¹ Line splitting and broadening observed for several residues in the 3D R³-CANCO spectra (e.g., K210, P33 and G212, Fig. 4) indicate irreversible changes in both loop regions (e.g., P33) and in the TM core (K210, G212) in the vicinity of exchanged sites. However, these changes remain localized and small, less than 0.8 ppm for CO, 1 ppm for N, and 0.4 ppm for C_α, and correspond to structurally similar substates formed after incubation at temperatures up to 76 °C.”

6) Page 13, Figure 5: *Retinal position is difficult to see, it would be better show the thicker bond for retinal in the figure.*

Response: We have changed the color to orange and used thicker lines to show retinal (Figure 6 and Supplementary Figure 6 in the resubmitted version).

7) Page 15, Results: *“The remaining ~60% population of retinal-bound ASR retain partially protected core in the reversibly unfolded state, which contribute to the 2D NCA spectrum (Fig. 2d) yielding the information about the partially unfolded intermediate.*

It is not clear the meaning of “partially unfolded intermediate”. Is that same as “reversibly unfolded state” and why this state is called intermediate?

Response: Thank you for the comment. We have revised our discussion on page 11 to eliminate any confusion, and also added a brief explanation on the meaning of an intermediate as follows (changes underlined):

“The remaining ~60% population of retinal-bound ASR retain partially protected core which contributes to the 2D NCA spectrum (**Fig. 2d**) yielding the information about the partially unfolded intermediate. This distinct intermediate is formed as a result of a sharp temperature-induced transition from mostly folded state observed at 76 °C, and is followed by an unfolded state formed at 83 °C. The latter state is characterized by nearly complete exchange of the backbone amides, and by a dramatic loss of color and retinal (Supplementary Figures 4h, 5g, 6g).”

8) Page 15, Results: *“Finally, incubation at 83 °C is accompanied by a dramatic loss of color; and results in nearly complete exchange of the backbone amides (Supplementary Fig. 4h, 5g, 6g).”*

It is not clear weather this state after incubation at 83 °C is reversibly unfolded state as same

as those appeared ~40% after incubate at 80 °C.

Response: Both states (or ensembles of states) contain no retinal (no absorption in the visible range). Unfortunately, we are unable to provide any specific structural detail because of the large inhomogeneous spectral broadening and lack of site-specific resolution.

9) Page 19, discussion: “ASR, in which we unveiled the unfolding pathway and mapped its stabilizing interaction. The observed unfolding intermediate state of ASR is trimer mainly helical (but with perturbed conformation of the seventh helix), and retains an intact cofactor pocket. Retinal serves as an interaction hub for the interior parts of helices by forming multiple non-covalent contacts with them and contributing to the stability of the helical bundle.”

In this discussion section, author may discuss more detailed unfolding pathways by including reversibly unfolding pathways after incubation temperature up to 76 °C, and irreversibly unfolding pathways which accompanied with losing retinal after incubation at 80 °C.

Response: We thank you for this suggestion. We added the following explanation on page 16 (changes underlined):

“Our approach was demonstrated using a well-studied seven-helical retinal binding membrane protein, ASR, in which we unraveled the unfolding pathway and mapped its stabilizing interactions. Unfolding begins in the TM core near retinal-binding K210, and appears to be associated with the formation of cavity near helix G. It gradually propagates towards the cytoplasmic side with increasing temperature, until a sharp transition to a distinct intermediate occurs in the ca. 76-80 °C range. The unfolding intermediate state of ASR is trimeric, mainly helical (but with perturbed conformation of the seventh helix), and it retains an intact cofactor pocket. Retinal serves as an interaction hub for the interior parts of helices by forming multiple non-covalent contacts with them and contributing to the stability of the helical bundle.”

We have also added some technical details on page 10:

“Progression of the exchange is observed in the loops and at the ends of helices upon incubation at higher temperatures up to 76 °C, along with general signal attenuation across the entire protein (Figs. 2c, 4, 5b, Supplementary Figures 4b-f, 5b-e). Although several residues in the TM core are exchanged in helices E, F, and G (I143, K167, T170-Y171, and S209-G212) after the incubation in the 48-68 °C temperature range, the TM core remains largely protected from solvent (Supplementary Figures 6b-e).”

Reviewer #3:

1. The authors emphasize much, rightfully so, the importance of reversibility in order to determine the stability of proteins, which particularly for membrane proteins represents a huge challenge. Contrary to what one might expect from the abstract and introduction, they have not demonstrated reversibility beyond the inflection point of the thermal denaturation curve which happens at 78C. That many features of the spectra might be reversible below this point, where

the protein is still relatively stable, is good and should be expected, but it does not prove reversibility of the actual unfolding reaction. This point must be clarified in the text.

Response. Thank you for the comment. The unfolding reaction is largely reversible up to 76 °C, partially reversible at 80 °C and irreversible at 83 °C (reported NMR temperatures). Although we state that solid state NMR allows monitoring the reversibility of the unfolding pathway, we cannot say much about the irreversibly misfolded state, other than that it is helical. Large inhomogeneous broadening prevents us from drawing more definitive conclusions.

We clarified in the abstract that we observe the reversible part of the unfolding pathway. We also say in the introduction that we follow unfolding below the irreversible denaturation point (page 4, last paragraph):

“Here, we employ thermally induced denaturation as the well-controlled external stimulus and detect H/D exchange by multi-dimensional MAS SSNMR to site-specifically follow gradual unfolding of a membrane protein as a function of increasing temperature below the irreversible denaturation point (**Fig. 1**).”

2. It would help much with the general understanding if the authors indicated in the quite important supplementary figure 2 with vertical lines the temperature points where their main NMR data were collected, i.e. 76 and 80 degrees for main figures and 48, 60, 68, 76, and 83 degrees for supplementary figures. BTW, the data of supplementary figures 4, 5, and 6 should be better referenced in the main text.

Response. We have added vertical lines in Supplementary Figure 2. We apologize for the omission of proper discussion of Supplementary Figures 4-6. We have added the following text on page 9-10 where these figures are referenced:

“We note that although solvent accessibility is expected to be the strongest factor contributing to the H/D exchange, hydrogen-bonding strength, local pH, protection of amide sites by adjacent sidechains⁴², as well as the time of solvent exposure (incubation for 1 hour in the 20-60 °C range, and for 2 minutes at above 60 °C to minimize the loss of retinal) are expected to contribute to its extent as well. Small peak attenuation (10-20%) observed for several residues in the TM regions may represent partial exchange occurring over the course of incubation. We confirmed that no significant additional exchange occurs during the 3D R³-CANCO experiments (~90 hours) by comparing 2D NCA spectra collected right before and after the 3D experiments.

Progression of the exchange is observed in the loops and at the ends of helices upon incubation at higher temperatures up to 76 °C, along with general signal attenuation across the entire protein (**Figs. 2c, 4, 5b, Supplementary Figures 4b-f, 5b-e**). Although several residues in the TM core are exchanged in helices E, F, and G (I143, K167, T170-Y171, and S209-G212) after the incubation in the 48-68 °C temperature range, the TM core remains largely protected from solvent (**Supplementary Figures 6b-e**).”

3. According to Online Methods, all experiments were performed on a single sample. Although

I realize that these types of experiments are demanding, demonstrating reproducibility is important, both for the NMR and DSC data. Some key experiments should be repeated with an independently prepared sample.

Response: We thank you for this suggestion. We have repeated key experiments on an independently prepared sample. DSC results are shown in **Supplementary Figure 2**. Small changes in the position of the temperature maximum (~1 °C) may be due to slightly different sample composition e.g., protein-lipid ratio, which was shown to have strong effects on the DSC results (e.g., ref. 26, Blume, A. Biological calorimetry: membranes. *Thermochim. Acta* **193**, 299–347 (1991)). For NMR, we followed the same H/D exchange and temperature incubation protocol, and collected additional NMR experiments including 1) reference 2D NCA experiment in H₂O; 2) 2D NCA experiment after incubation in D₂O at 20 °C; 3) 2D NCA experiment after incubation in D₂O at 76 °C; 4) 2D NCA experiment after incubation in D₂O at 80 °C; 5) reference 2D carbon-carbon correlation spectrum in H₂O; 6) 2D carbon-carbon correlation spectra after incubation at 20 °C, 76 °C, 80 °C. These additional experiments demonstrated the overall similar unfolding pattern; they are described in the main text (pages 7 and 12) and are included in the Supplementary Materials as Supplementary Figures 10 and 11. Technical details are described in the Methods.

Page 7:

“Two independently prepared but otherwise similar ASR samples were used in this study. A larger sample (sample 1, ~10 mg of ASR) was used to record 2D NCA and 3D R³-CANCO spectra, and sample 2 was used to validate our conclusions.”

Page 12:

“To validate our results and ensure their reproducibility, we have conducted an additional set of experiments on an independently prepared ASR sample (sample 2). Overall, similar behavior was observed for this sample (**Supplementary Figures 10, 11**). ASR undergoes largely reversible unfolding after incubation at 76 °C prior to the major unfolding event, as evident from similar 2D NCA peak patterns (**Supplementary Figure 10c**) and from conserved chemical shifts in the 2D ¹³C-¹³C correlation spectrum (**Supplementary Figure 11a**). Unfolding becomes partially irreversible at 80 °C: the preserved, correctly folded core represented by the cross peaks in the 2D NCA spectrum is similar to that of sample 1 (**Supplementary Figure 10d**), whereas irreversibly unfolded parts of the protein result in broad peaks in the 2D ¹³C-¹³C correlation spectrum (**Supplementary Figure 11b**). Small discrepancies between samples 1 and 2, e.g., appearance of a cross peak corresponding to A13 in sample 2 at 80 °C, may be due to the different amount of protein in sample 2, or slight variations in sample incubation conditions.”

Supplementary Figure 10. A series of 2D NCA correlation spectra as a function of incubation temperature collected on independently prepared sample 2. (a) Reference 2D NCA spectrum collected in the H₂O based buffer. (b) 2D NCA spectrum collected after incubation in the D₂O based buffer at 20 °C. (c to d) Overlay of 2D NCA spectra collected on sample 1 (red) and sample 2 (black) after incubation at 76 °C (c), and at 80 °C (d).

Supplementary Figure 11. 2D DARR ^{13}C - ^{13}C correlation spectra collected on sample 2. (a) A comparison between the 2D ^{13}C - ^{13}C correlation spectrum collected before incubations (blue) and after incubation in D_2O at 76 °C (red). **(b)** A comparison between the 2D ^{13}C - ^{13}C correlation spectrum collected before incubations (blue) and after incubation in D_2O at 80 °C (red). DARR mixing was 30 ms in all experiments.

4. Related to the previous point, the lipids for bilayer reconstitution were DMPC:DMPA (9:1), but there was also some TX100 detergent needed in the $\text{}$. Was the detergent fully removed by 24h Bio-bead treatment? How much might have been left in the sample and how could this affect the unfolding data? It is important to show that the sample preparation procedure and results are reproducible and therefore support the overall conclusions of the paper.

Response: Triton-mediated lipid reconstitution with Bio-beads detergent removal is a well-established procedure for rhodopsins (e.g., Pitard et al., Eur J Biochem. 235 (1996), 769; Rigaud et al., BBA 1231 (1995) 223), employed by us in many successful studies of ASR and proteorhodopsin (e.g., Shi et al., J. Mol. Biol., 386 (2009) 1078; Shi et al., Angew Chem. 50 (2011) 1302). We have developed a number of controls to verify complete elimination of the detergents (both Triton and DDM). A simple foaming test is usually sufficient (if the detergent is not removed, the liposome suspension will foam upon shaking). We also perform an additional verification by FTIR by monitoring Triton X-100 absorption bands (the strongest one around 1113 cm^{-1}). It was not present in either of the two samples.

We added the following sentence on page 18:

“The removal of Triton was confirmed by the disappearance of the Triton X-100 absorption bands in the FTIR spectra”

5. (minor) Some of the figure panels are rather small (e.g. figure 3a-c) and hard to read.

Response: We have revised figures and split figure 3 into two figures (3 and 4) for better legibility.